# The Embodiment Gap in Robot Foundation Models

## Abstract

Robot foundation models (RFMs), including vision-language-action (VLA) policies, are often discussed through a scaling view: more data, larger models, and broader benchmarks should improve generalization. In robotics, however, a model can generalize while work still remains before it can run on a robot with a particular body. The work required differs across methods and target robots, and those differences affect practical deployment. We call the gap between reusable models, representations, or data and their use in execution on the target robot the *embodiment gap*. This survey examines what can be reused across robot embodiments and what must still be implemented on a new robot. We place existing methods on a two-axis map that shows the type of shared structure and the stage at which adaptation is needed for execution on the target robot. We then examine recent work through three overlapping research directions: sharing semantics and perception, sharing robot data and interfaces, and learning correspondence across embodiments. We also propose a reporting framework for adaptation work that success rate alone does not reveal. The framework identifies the work that should be checked when comparing cross-embodiment learning and highlights work that remains on a new robot and questions for future study.

## 1 Introduction

The scaling ideas that shaped language and vision are increasingly influencing robot learning. Large models are trained on diverse data with the aim of obtaining broad generalization. Robot foundation models, especially VLA policies and generalist robot policies, represent this direction. RT-X, Octo, OpenVLA, RoboCat, and the $\pi_0$ family are prominent examples (Octo Model Team et al., 2024; Kim et al., 2025c; Open X-Embodiment Collaboration et al., 2023; Khazatsky et al., 2024; Zhao et al., 2023; Brohan et al., 2023b;a; Driess et al., 2023; Bousmalis et al., 2024; Black et al., 2024; Physical Intelligence et al., 2025).

Recent work increasingly seeks to reuse knowledge, representations, and experience across individual robots. A robot, however, ultimately acts through a particular body. A shared model or representation does not directly become motion on a real robot. It must be connected to the body and control system of the target robot so that the robot can execute it. We call the gap between reusable structure and execution on a particular robot the *embodiment gap*.

Many robot foundation models are evaluated mainly by whether the robot eventually succeeds. The practical value of two systems can still differ when they require different amounts or kinds of work before reaching the same success rate. This issue has long affected robot deployment. In industrial robotics, for example, system integrators often carry out much of this less visible work through experience and tacit knowledge. The same issue appears when a robot foundation model is deployed on a new body, often under new deployment conditions. Final performance therefore needs to be considered together with the work required to reach deployment.

Recent discussions about scaling robot data and models reinforce this concern. Goldberg argues that language and vision can draw on Internet-scale training data, whereas robotics cannot obtain observations paired with action commands at a similar scale. He further argues that useful robot data will accumulate only

---

The authors used GPT-5.5 to proofread the manuscript and prepare a draft of Figure 1. The authors take full responsibility for the final manuscript.

after learning methods and engineering methods are combined well enough to make robots operate reliably (Goldberg, 2025). A related debate in *Science Robotics* also treats the roles of data, models, and engineering in robotics as an open question (Amato et al., 2025). We still lack a clear account of which gaps between a shared model and execution on a target robot can be reduced by scaling data or models, and which remain as engineering work on that robot. This survey studies how different directions for scaling data and models change the embodiment gap and the work that remains on the target robot.

We organize recent work into three overlapping research directions and examine how each direction connects shared information to execution on a target robot. The first direction shares semantics and perception. The second shares robot data and interfaces. The third learns correspondence across embodiments. We also propose a reporting framework for the adaptation process behind a reported success rate. The framework provides a way to examine a system's result on a new robot together with the work required to obtain that result.

Recent surveys cover foundation models and VLA policies in robotics, robot data, benchmarks, and evaluation methods (Firoozi et al., 2025; Hu et al., 2023; Zhong et al., 2025; Kawaharazuka et al., 2025; Motoda et al., 2025; McCarthy et al., 2025; Zheng et al., 2026b; Wang et al., 2026; Gao et al., 2025; Raychaudhuri and Chang, 2025). Building on that literature, we focus on where the embodiment gap appears in robot foundation models for manipulation, especially VLA policies and generalist robot policies.

We also include mechanisms around the model that make shared models or representations executable on a specific target robot. These include the design of data and interfaces, methods that account for differences between robot bodies, and work on real-robot execution and evaluation. Such work is not always itself a robot foundation model. It is nevertheless central to running a shared model on a target robot and to understanding the embodiment gap. Appendix A explains the survey scope, literature-selection strategy, boundary cases, and our treatment of recent preprints and OpenReview submissions.

This survey makes three contributions. First, we define the embodiment gap as the gap between reusable models, representations, or data and their use in execution on a particular robot. Second, we place representative methods on a two-axis qualitative map and use it to describe the strengths and limits of three research directions. Third, we propose a reporting framework for recording adaptation work and failure causes on the target robot.

The paper is organized as follows. Section 2 explains the embodiment gap and the adaptation work that remains on the target robot. Section 3 introduces the two-axis map and the three research directions. Sections 4–6 examine the three directions in turn. Section 7 presents ways to report adaptation work that success rate alone does not reveal. Section 8 discusses scaling with the embodiment gap in view, and Section 9 concludes. The appendices provide details on literature selection, adjacent concepts, placement rationales, and a compact reporting checklist.

## 2 The Embodiment Gap in Robot Foundation Models

### 2.1 Operational definition of the embodiment gap

In this paper, the embodiment gap is the gap that arises when models, representations, or data reused across robots must be converted into executable actions that match the body and control system of a target robot. We include a problem in this gap when additional work is required on the target robot after transfer across embodiments. This idea is related to the broader literature on transfer learning (Taylor and Stone, 2009; Pan and Yang, 2010), but our scope is narrower: execution across robot embodiments. We use *shared structure* as an umbrella term for models, representations, and data that are reused across robots. Some shared structures, such as task meaning and plans, are far from execution. Others, such as action representations and skills, are closer to execution. Figure 1 gives a conceptual overview.

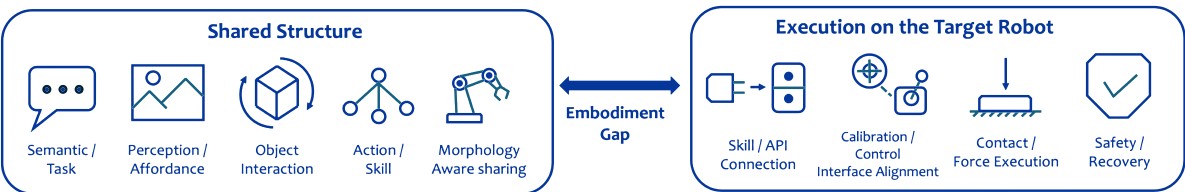

Figure 1: Conceptual overview of the embodiment gap. The left side shows structures reused across robots: semantics and tasks, perception and affordances, object interaction, actions and skills, and morphology-aware sharing. The right side shows work that often remains when these structures are executed on a target robot: skill or API connection, calibration and control-interface alignment, contact and force execution, and safety and recovery. The embodiment gap arises in moving from shared structure to execution on the target robot.

High-level plans and meaning are often reusable across many robots, but several steps are still needed before they become physical motion. Action representations and skills are closer to execution, yet they depend more strongly on the kinematics and control system of the target robot. These differences include engineering work that is often difficult to see in a paper. Section 3 organizes them in a two-axis map and uses the map to position existing methods.

## 2.2   Relation to adjacent concepts and contrasting examples

The definition above gives a practical test for inclusion: does the problem arise when a shared model or representation is executed across different robot embodiments? Changes in appearance, task, sim-to-real conditions on the same robot (Zhao et al., 2020; Tobin et al., 2017), and general integration work are all important. They are outside the main scope of this survey when they can occur without a change in robot embodiment. For example, a change in lighting or object appearance on the same robot is mainly a domain-shift problem (Csurka, 2017). Learning a new procedure on the same robot is mainly task adaptation (Taylor and Stone, 2009; Pan and Yang, 2010).

These issues become part of the embodiment gap when they are tied to deployment on another robot. One example is using the same policy on a different arm and having to adjust the control system or action interface (Lum et al., 2025). Even when the goal remains "lift the cup," a change in the number of fingers on the end effector can change the grasp and the contact pattern needed to succeed.

The boundary with system integration also needs to be clear. A robot system always requires hardware connections, configuration, and implementation. Our focus is the part of that work that makes a shared policy or representation executable on a new robot. It includes adjusting motion to the target robot, making execution safe, and enabling the system to recover after failure. This work is often difficult to see in performance reports for robot foundation models, yet it matters directly to the embodiment gap.

Action-space mismatch is a common example. Different robots may expose different policy outputs or control interfaces, so a shared representation cannot always be executed directly. Even when two systems use Cartesian end-effector commands, the same displacement does not necessarily produce the same physical behavior. A different gripper or mobile base can change how the robot makes contact with an object and how it recovers after an error.

For this reason, the survey covers the full set of tasks needed to run shared structure on a target robot, including work that goes beyond a simple difference in action format. Appendix B presents contrasting inclusion and exclusion examples for adjacent concepts.

## 3   Where and Why Does the Embodiment Gap Appear?

This section introduces a two-axis qualitative map for comparing existing work.

### 3.1 Construction of the two-axis map

The horizontal axis orders shared structure by its distance from execution on the target robot. At the farthest end are semantics and tasks, which describe goals and plans. Next come perception and affordances, which identify objects and places for action, followed by object interaction, which describes how an object or scene should change through manipulation. Actions and skills are closer to execution. The closest category is morphology-aware sharing, where information about the robot body is supplied to the model. This ordering draws on prior surveys of robot foundation models, VLA policies, learning control from video, and evaluation of generalist robot policies (Firoozi et al., 2025; Zhong et al., 2025; McCarthy et al., 2025; Gao et al., 2025).

Moving to the right brings the shared information closer to execution. Meaning and plans can often be reused across robots, but they do not determine a physical motion. Object motion and action representations provide more direct guidance, while also depending more strongly on the body and control system of the target robot. The methods at the far right provide joint structure or kinematic information so that a policy can account for body differences within the model (Sferrazza et al., 2024; Patel and Song, 2025; Zheng et al., 2026a; Suzuki et al., 2026; Zhang et al., 2026).

A system may combine several kinds of shared structure. In such cases, we choose the structure that supports the system's main claim about transfer across embodiments. A method that mainly indicates where a robot should act is placed under perception and affordance. A method that mainly describes how an object or scene should change is placed under object interaction. For methods such as action motifs that span several stages, we examine what the method ultimately passes to the target robot (Xu et al., 2025; Bharadhwaj et al., 2024; Zhi et al., 2026). Some method groups primarily contribute data infrastructure. Because the horizontal axis has no separate category for data infrastructure, we place them under action and skill when their trajectories or actions are reused as a common training resource. Appendix C records this forced choice and the rationale for each placement.

The vertical axis shows the stage at which the main work remains when shared structure is executed on a target robot. The upper stage connects an abstract plan to skills the robot can call. The next stage aligns model outputs with the target robot's control interface so that the commands have the intended physical meaning. Closer to physical execution, the robot must establish stable contact and correct deviations while contact is maintained. The final stage concerns safe stopping, retrying, and recovery when failures or disturbances occur. These problems extend beyond robot foundation models and remain difficult throughout real-world robotics. The stages closer to execution still require substantial adjustment and testing on real hardware.

Some methods leave work at both control-interface alignment and contact-rich execution. We place such a method according to the work that must be completed first for execution on the target robot. If the method still needs a robot-specific action decoder, an execution policy, or additional target-robot training, we place it under calibration and control-interface alignment. If the method already maps its representation to target-robot actions and the remaining difficulty is explicitly associated with grasping, force, friction, contact dynamics, or closed-loop physical correction, we place it under contact and force execution.

The two axes are useful because the type of representation alone does not show how far that representation carries a system toward execution on a target robot. A representation hierarchy can separate high-level planning methods from action-representation methods. It does not show the work required to connect a plan to available skills or to align an action representation with a target robot's controller. An analysis of invariance across embodiments has a related limitation: it does not show whether the system can establish contact or recover after failure.

Figure 2 is not used to predict a numerical amount of work or to rank methods. It places representative work qualitatively and descriptively so that readers can see where problems are likely to arise and what adjustments or failures should be reported. Methods that share meaning and plans often need a connection to executable skills. Methods that share action representations often need the outputs to be aligned with the target robot's motion. Methods that share structures closer to execution are more directly affected by contact and recovery. These observations also connect the map to the reporting items and failure causes discussed in Section 7.

## 3.2 A qualitative map of existing work

Figure 2 places representative work on the two axes defined in Section 3.1. We include robot foundation models and generalist policies as well as work that supports data collection and real-robot deployment. Three researchers independently coded 21 systems or method groups using a shared initial codebook described in Appendix C. For each group, they selected one primary category on each axis and recorded a secondary candidate, confidence, and a boundary-case flag when the choice was uncertain.

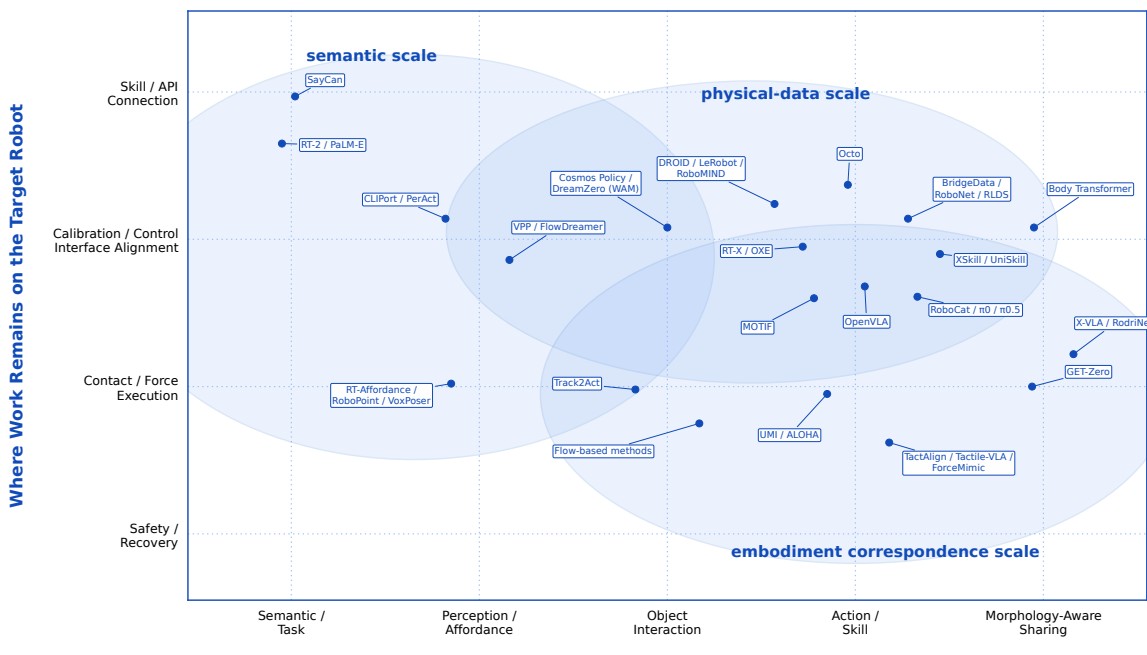

Figure 2: Two-axis qualitative map of the embodiment gap. The horizontal axis shows what is primarily shared across robots. The vertical axis shows where work primarily remains when that shared structure is executed on the target robot. Each point is the final placement after independent coding by three researchers and adjudication of boundary cases. The map does not estimate the amount of work or rank methods. Within-row shifts show boundary cases and avoid label overlap; the categories in Table 7 are unchanged. No method group was assigned to the Safety / Recovery row after adjudication. The translucent regions show the three research directions discussed in Sections 4–6. The placements in Figure 2 and Table 7 come from the same coordinate table.

On the shared-structure axis, all three coders agreed on 16 groups (76.2%), and at least two coders agreed on all 21 groups (Fleiss' $\kappa = 0.78$). On the axis for where work remains, all three agreed on 10 groups (47.6%), and at least two agreed on 20 groups (95.2%; $\kappa = 0.33$). The coders agreed on both coordinates for 7 groups (33.3%). Ten of the 11 disagreements on the vertical axis were between adjacent categories. Most concerned whether the primary remaining work was alignment of a policy or control interface with the target robot, or the contact-rich execution that follows. XSkill / UniSkill was the only group for which the three coders selected three different stages.

These results indicate that the shared-structure axis can be coded relatively consistently, while the boundary between control-interface alignment and subsequent contact execution requires judgment. We preserved the independent records and adjudicated the boundary cases after rereading the main claims of each method and the target-robot work identifiable from public materials. The independent check showed that the boundary between calibration and contact caused most of the disagreement, so we added the priority rule in Section 3.1 during adjudication. The final coordinates are stored in a single machine-readable table, which provides the

placements used in both Figure 2 and Table 7. Appendix C gives the full procedure, boundary rules, and placement rationales.

Figure 2 also reveals a relationship between shared structure and work on the target robot. Work tends to move closer to physical execution as the shared structure itself moves closer to execution, but the relationship is not one-to-one. OpenVLA and UMI / ALOHA, for example, both share actions or skills. OpenVLA still requires its outputs to be aligned with the target robot's controller, whereas UMI / ALOHA leaves the problem of reproducing collected motion as stable contact on the target robot. The remaining work therefore depends on how far the method's design extends into execution on the target robot. Section 3.3 discusses this design choice.

None of the 21 method groups was placed in the Safety / Recovery row. This finding does not mean that safety and recovery have been solved. Within the scope of this survey, the ability to stop safely after failure and resume or recover has not yet become a central research target of robot foundation models. The issue concerns the whole system, including the model, control, and system integration. One possible architecture would use a generalist policy to generate motion while a higher-level agent decides when to stop or recover. Whatever architecture is chosen, a high success rate does not by itself show whether the system can operate continuously in the real world when the stopping and recovery process is unclear. Figure 2 therefore exposes research gaps as well as category boundaries and helps identify the work and failures that future papers should report.

### 3.3 Three scaling directions and why the gap remains

The placements in Figure 2 reflect a design choice about what each method attempts to scale. Most methods do not try to address the entire embodiment gap at once. They first select a structure that can be shared and then scale that structure. This is a reasonable engineering choice. It also determines what the method handles well and what additional adjustment remains on the target robot. Figure 3 summarizes these design choices as three overlapping research directions.

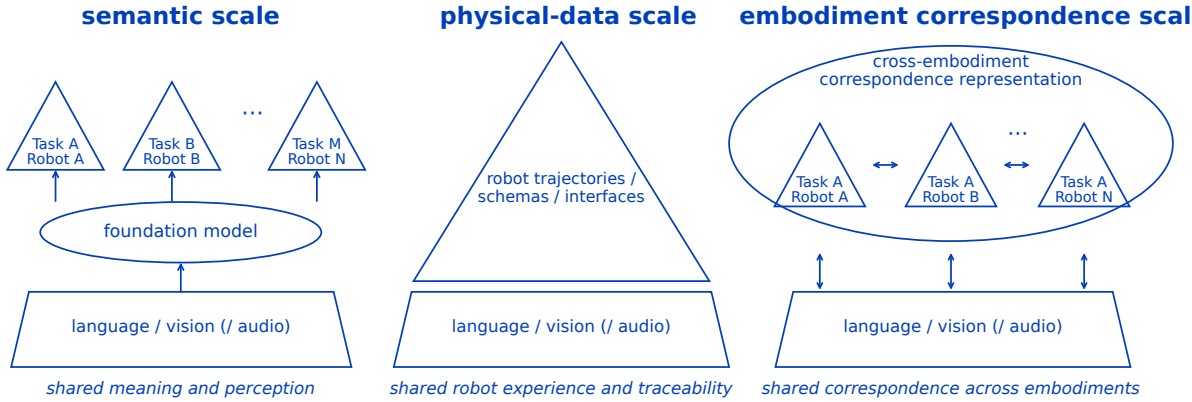

Figure 3: Three research directions viewed through the embodiment gap. Sharing semantics and perception broadens the meanings and visual cues available to a robot. Sharing robot data and interfaces makes robot experience easier to reuse and its conditions easier to trace. Learning correspondence across embodiments models relations among different bodies through object motion, abstract actions, morphology, and contact cues. The directions overlap and are not mutually exclusive categories.

The first direction shares semantics and perception. It is closely connected to foundation models trained on the large amount of language, image, and video data available on the Internet. This direction can use data at scale and share broad information about what a robot should do or where it should attend. Robot planning and visual understanding can then use this information downstream. The main learning effort, however, is directed toward making semantics and perception broadly reusable. Work is still needed to turn those cues

into stable action on the target robot. Video foundation models and world models may extend this direction toward structures closer to execution because they can represent change over time in objects and scenes.

The second direction shares robot data and interfaces. Physical robot data are more difficult to collect than language or image data. It is therefore reasonable to standardize some aspects of the robot body, the way data are collected, or the way actions are represented so that physical data can be reused. Deploying many robots with the same body, for example, makes it easier to train a large policy for that body. When bodies cannot be standardized completely, common interfaces such as UMI and common trajectory or action schemas can still make data easier to share. This direction obtains scale mainly through shared data formats and interfaces. Adjustment on the target robot and operation during evaluation often remain necessary.

The third direction learns correspondence across embodiments. It seeks to model which aspects of behavior can be related between robots with different body structures. In the longer term, this direction could lead to RFMs that generate actions across several embodiments while accounting internally for their differences. The field has not yet established which data should be collected or which representations should be scaled for that goal. Current work explores correspondence through object motion, abstract actions, morphology, and contact cues.

When the three directions are overlaid on Figure 2, they occupy different regions. Work that shares semantics and perception often lies farther from execution, although video foundation models and world models extend toward object motion and temporal change. Work that shares robot data and interfaces spreads across the middle of the map. It uses physical robot data and is therefore closer to execution, but scaling such data is still difficult, and substantial work remains in control and contact. Work on correspondence across embodiments has the potential to share structures closest to execution, while its data and representations are still at an early stage of scaling.

The overall picture is that RFMs are expanding what can be shared while still leaving substantial adjustment for execution on a target robot. Future systems need to incorporate more of that adjustment into the model or system design. Papers also need to state where human work was required during real-robot deployment. Sections 4–6 examine how the design of each research direction creates its strengths and why execution problems remain.

## 4 Sharing Semantics and Perception

This research direction brings knowledge from large language and vision datasets into robotics. Its main strength is that task meaning, object information, and spatial cues can be reused across robots. A robot can therefore reason about what to do and where to attend without learning everything from data collected on that particular robot. Methods at the semantic and task level share high-level information about what should be achieved, leaving detailed robot motion to the execution system. SayCan, for example, matches action candidates proposed by a language model to skills that the target robot can execute (Ahn et al., 2023). Related work uses large language models as interfaces for planning or program generation (Liang et al., 2023; Singh et al., 2023; Liu et al., 2023a). These methods help decompose tasks and select skills. The language model mainly produces a plan or a set of candidate actions, while the executable skills must already exist on the target robot. A sound plan cannot be executed when the required skills are missing. Semantic understanding therefore still needs a separate connection to the robot's executable skills.

Methods based on perception and affordance bring shared semantics one step closer to execution. They provide cues about which object the robot should act on and where it should make contact. CLIPort and PerAct connect language and vision to spatial representations for manipulation (Shridhar et al., 2022; 2023). R3M, RT-Affordance, RoboPoint, and VoxPoser also provide visual or spatial cues about objects and possible actions (Nair et al., 2023; Nasiriany et al., 2024a; Yuan et al., 2025; Huang et al., 2023). Perception Stitching separates a transferable visual encoder from the subsequent visuomotor execution (Jian et al., 2024). These methods can identify where a robot should act. They do not determine how the target robot should approach that location or maintain contact after touching the object. A change in reachability or gripper shape can turn the same visual cue into a different motion. As ActionEQA shows, semantic instructions increasingly expose geometric and physical reasoning failures as they move toward low-level action (Bao et al., 2026).

Video and world-model methods provide cues that are closer to execution because they represent how a scene changes over time. Predicting changes in objects or in a human demonstration can narrow the range of actions that a target robot needs to consider. This idea appears in work on visual foresight, video prediction, affordance estimation, and imitation from human video (Nasiriany et al., 2024a; Ebert et al., 2018; Hu et al., 2024; Wu et al., 2023; Yuan et al., 2025; Guo et al., 2025; Huang et al., 2023; Wang et al., 2023; Kareer et al., 2024; Hoque et al., 2025). Sharing temporal change brings semantics and perception closer to action generation. Current video models mainly provide a desired change or a candidate action. The motion that realizes the change on a particular robot and the correction needed when contact drifts still have to be designed. This direction can move closer to execution if temporal knowledge from video is connected to closed-loop control on real robots.

Recent VLA policies and generalist robot policies combine shared semantics and perception with shared robot data. RT-2, PaLM-E, RT-X, Octo, OpenVLA, RoboCat, and the $\pi_0$ family typically use pretrained vision-language models together with multi-robot datasets to train large policy backbones on robot data (Octo Model Team et al., 2024; Kim et al., 2025c; Open X-Embodiment Collaboration et al., 2023; Khazatsky et al., 2024; Zhao et al., 2023; Brohan et al., 2023b;a; Driess et al., 2023; Bousmalis et al., 2024; Black et al., 2024; Physical Intelligence et al., 2025). This is an important step toward turning semantic and visual cues into actions. The action representation produced by a VLA policy still reflects the robots and control conventions in its training data. If the target controller interprets the same output with a different speed or magnitude, the same action token can produce different motion and contact. Studies of VLA tuning and fine-tuning show that target-robot data are often needed to align this relationship (Zhang et al., 2025b; Kim et al., 2025b). Reported success rates should therefore be read together with the adjustment and real-robot testing performed on the target robot.

High-level decisions from this direction combine naturally with task-specific motion-generation policies and existing control systems. Much of the learning effort, however, is concentrated on producing plans or visual cues. Turning a cue into motion requires connecting the plan to skills available on the target robot and aligning the selected skill's output with that robot's motion. Once the robot touches an object, it may also need to correct the motion to maintain stable contact. These steps often lie outside the main learning target of models for semantics and perception. The central remaining problem is therefore the connection between high-level decisions and stable execution on the target robot.

## 5 Sharing Robot Data and Interfaces

This research direction turns difficult-to-collect physical data into training resources that can be reused across robots. It seeks common ways to record robot experience and common interfaces for collecting it, so that data from different sources can train the same model. Large robot datasets and data infrastructures have changed how generalist robot policies are trained. RoboNet, BridgeData V2, Open X-Embodiment / RT-X, DROID, RLDS, LeRobot, RoboMIND, OXE-AugE, and RoboWheel treat robot experience as a shared training resource (Open X-Embodiment Collaboration et al., 2023; Khazatsky et al., 2024; Ramos et al., 2021; Cadène et al., 2026; Walke et al., 2023; Dasari et al., 2020; Wu et al., 2025; Ji et al., 2025; Zhang et al., 2025c). Public and well-documented datasets make it easier to combine experience across studies. They can also make the conditions under which the data were collected easier to inspect. For example, it matters whether a policy output described end-effector motion or joint motion, how frequently commands were sent, and how the camera was aligned with the robot. Such information helps explain why the same model may behave differently on another real system. Without it, readers cannot easily separate a failure of the model from a mismatch in execution conditions. The level of detail recorded by current datasets still varies considerably.

Teleoperation and data-collection interfaces also shape the physical data that can be shared. The ALOHA family provides low-cost bimanual teleoperation for fine manipulation (Zhao et al., 2023; Fu et al., 2025; ALOHA 2 Team et al., 2024). DROID moves toward large-scale in-the-wild manipulation data collected with standardized procedures and metadata (Khazatsky et al., 2024). UMI, OPEN TEACH, and AnyTeleop show that the way human operation is recorded influences later transfer and execution (Chi et al., 2024; Iyer et al., 2025; Qin et al., 2023). These interfaces make it easier to record human operation in a consistent form

and to collect many examples. Each interface also assumes a particular gripper, input device, or pattern of robot motion. Replaying an end-effector trajectory on a robot with a different arm length or gripper can place the goal out of reach, cause a collision, or change the way the robot touches the object. The recorded motion must therefore be adjusted so that it remains feasible and safe on the new body. A widely deployable common interface can still support large-scale collection under a shared operation format. Its value and its dependence on a particular embodiment need to be understood together.

Data schemas and normalization help place heterogeneous data into a common training format. RLDS represents datasets as episodes and steps (Ramos et al., 2021); Open X-Embodiment combines multiple datasets under common conventions (Open X-Embodiment Collaboration et al., 2023); and LeRobot connects a shared data format to tools for training and deployment (Cadène et al., 2026). Actions can also be represented relative to an object or task, which reduces direct dependence on robot-specific joint coordinates. Object-centered frames and task frames describe end-effector or object motion in relation to the task. End-effector poses and SE(3) transforms describe changes in position and orientation geometrically. Equivariant representations are designed to change consistently when position or orientation changes. These choices can make the intended motion easier to compare when robot shape or installation differs (Wang et al., 2019; Pan et al., 2023; Liu et al., 2023c; Zhang et al., 2025d; Yang et al., 2023a). Action tokenization and other common action representations also make heterogeneous data easier to train on jointly. A common action representation does not guarantee common physical execution. The instruction to move an end effector slightly forward, for example, can produce a different amount of contact or pressure on another robot. The difference may be small on a simple task. It can lead to slipping, insufficient insertion, or excessive force when contact is central to success. Shared action formats therefore remain separate from the physical question of whether the action succeeds on a different body and controller.

Diffusion Policy and related trajectory-generating policies can produce smooth trajectories and several plausible action candidates from the distribution in the training data (Chi et al., 2023; Hsu et al., 2025; Bauer et al., 2025). This is useful when manipulation admits several valid strategies or when a longer motion should be generated as a coherent sequence. These policies learn how observations and robot commands corresponded in the training data. When the camera placement or robot response changes at deployment time, a similar generated trajectory can lead to a different speed or pattern of contact. A trajectory that looks natural in the data may therefore be unstable on the real target robot. The main challenge is to connect generated trajectories correctly to the target robot's observations and closed-loop control.

Frameworks such as PyRobot and LeRobot make it easier to use common procedures for loading data, training a policy, and deploying it on a robot (Cadène et al., 2026; Murali et al., 2019; Jülg et al., 2026; Zhu et al., 2020; Kumar et al., 2023). They reduce the amount of software that each project must rebuild. The same software command still does not produce the same physical behavior when robots use different controllers or camera placements. Robots also differ in reachability and in how they interact with surrounding objects. A framework can standardize the software connection, while physical alignment on the real robot still requires separate work.

Benchmarks such as LIBERO and RLBench make methods easier to compare by standardizing tasks and success criteria (Liu et al., 2023b; James et al., 2020; Mees et al., 2022; Nasiriany et al., 2024b; Heo et al., 2023; Parakh et al., 2025; Zhou et al., 2025; Liu et al., 2026). Existing evaluations, however, usually emphasize final success rate and often provide little detail about the work required before real-robot evaluation could begin. A reader may not know how often the camera or coordinate frame was readjusted, how often a person stopped the robot, or how much manual resetting followed a failure. Without this information, similar success rates do not establish similar deployment conditions. Future benchmarks should record the process that made performance measurement possible as well as the final performance.

Compared with methods that share only semantics or perception, this direction works with data that are closer to robot action. A common format can also support the training of larger policies and make experimental conditions easier to trace. The same command still need not produce the same motion after the body or controller changes. The main remaining work is therefore to connect shared data and policies to the physical behavior of the target robot. Small differences in that connection can determine success or failure in contact-rich manipulation.

# 6 Learning Correspondence Across Embodiments

This research direction studies how experience obtained with one body can be converted into a form that another body can use. Different methods use different cues to relate embodiments. Some share the stage of task progress. Others describe how an object should move. Methods closer to execution provide the model with information about the robot body or use signals measured during contact. These choices represent correspondence at different levels of abstraction, and they leave different problems on the target robot. High-level correspondence relates behavior through task meaning or progress and does not require a direct match between detailed robot motions. Sharing what should be achieved at each stage and where the interaction should occur can relate behavior across embodiments. XSkill and UniSkill learn skill representations with meaning across bodies (Xu et al., 2023; Kim et al., 2025a). Methods based on human video extract the location of interaction or task intent from a demonstration and use those cues for transfer (Nasiriany et al., 2024a; Bahl et al., 2023; Srirama et al., 2024; Chen et al., 2026). Mirage, SHADOW, and RoVi-Aug reduce visual differences between bodies to support transfer (Chen et al., 2024; Lepert et al., 2025; Chen et al., 2025b). A high-level correspondence can still leave the target robot with the tasks of turning that intent into safe motion, maintaining contact, and recovering after failure.

Intermediate correspondence shares how an object or scene should change without requiring a direct match between robot joints. Describing where an object should move or which pose a tool should reach makes it easier to relate the intended outcome across different robots. This idea appears in object flow and point tracking, object-centered motion, and tool-centered representations (Xu et al., 2025; Zhi et al., 2025; Chen et al., 2025c; Yin et al., 2025; Hsu et al., 2025; Chen et al., 2025a). Track2Act uses point motion from video to specify where an object should move (Bharadhwaj et al., 2024). The object goal does not determine where the target robot should place its gripper or how it should maintain contact. A different gripper shape or control response can cause the object to slip or prevent the robot from applying enough pressure, even when the desired object motion is the same. The point tracks therefore still need a target-robot policy and a mechanism for correcting deviations during contact.

The video and world-model methods discussed in Section 4 can also provide cues for correspondence because video contains temporal change and object motion. VPP and FlowDreamer use representations that predict changes in a scene (Hu et al., 2024; Guo et al., 2025). Cosmos Policy adapts a pretrained video model to robot control (Kim et al., 2026). DreamZero treats future world states and actions together in a World Action Model and studies transfer from video-only demonstrations and few-shot adaptation (Ye et al., 2026). Video and world models describe how a scene changes over time through an action. In this sense they are closer to execution than methods based only on a static image. The video mainly specifies a desired change, while each robot still needs a motion that can produce that change. Deployment requires converting the prediction into the robot's reachable space and aligning the timing and speed of contact. A motion that looks natural in video can otherwise contact the object at the wrong place or apply a different force on the real robot. Connecting temporal knowledge from video to closed-loop control on the target robot is an important open problem.

The methods above mainly use changes in objects or scenes. Latent-action and action-motif methods represent the action at an intermediate level. Their representations describe task progress or action intent and avoid direct sharing of low-level robot commands (Bauer et al., 2025; Ye et al., 2025; Bu et al., 2025; Zha et al., 2026; Huang et al., 2026a). MOTIF learns action motifs from data collected on different robots and uses them for few-shot cross-embodiment transfer (Zhi et al., 2026). This design can share action structure that is close to execution while avoiding direct dependence on robot-specific commands. It still does not determine the contact produced by the abstract action on a target robot. Real-robot use needs closed-loop correction based on the target robot's observations and a way to recover when contact deviates.

Methods closer to execution provide the policy with the robot's body structure. Body Transformer and GET-Zero describe how joints, sensors, and actuators are connected so that action generation can depend on the embodiment (Sferrazza et al., 2024; Patel and Song, 2025). X-VLA, morphology-aware Transformers, and RodriNet use body-specific information or kinematic relations to align policy outputs with the target robot (Zheng et al., 2026a; Suzuki et al., 2026; Zhang et al., 2026). Related studies also bring body differences into the policy (Chen et al., 2018; Wang et al., 2018; Gupta et al., 2022; Hu et al., 2022; Xiong et al., 2023; Wei

et al., 2024; Xiong et al., 2024; Przystupa et al., 2025; Wang et al., 2024b; Wu et al., 2026). Morphology-aware methods can reduce simple motion mismatch caused by different arm lengths or joint structures. Knowing the kinematic structure does not determine how a real gripper presses against an object or how much it slips. Contact depends on the end-effector shape and material as well as the response of the real controller. Contact adjustment and recovery therefore remain important even when the body structure is represented explicitly.

Force and tactile information become important when a method addresses contact directly. TactAlign and UniTacHand seek to align tactile observations across embodiments. Feel the Force and ForceMimic extract cues about force and contact from human motion or demonstrations that do not use the target robot (Wi et al., 2026; Zhang et al., 2025a; Adeniji et al., 2025; Liu et al., 2024). Tactile-VLA and TaF-VLA show how tactile or force signals may be used as inputs or alignment signals for VLA policies (Huang et al., 2025; 2026b). Work on tactile perception also suggests that touch can compensate for capability differences across embodiments (van den Bogert et al., 2024). These methods make contact deviations observable when vision alone is insufficient. The meaning of a tactile signal still depends on sensor placement and on how its values are measured. The same signal magnitude on another robot may describe a different contact state. A deployed system must therefore align the signal with the real robot and make tactile observations comparable across embodiments. Touch can make contact easier to observe while creating new adaptation work for sharing those observations. With sufficient data, future systems may learn contact or force representations that remain useful across these differences.

The correspondence data discussed in Section 5 also support this direction. UMI uses a portable gripper and policy interface to make in-the-wild human demonstrations easier to use for robot policies (Chi et al., 2024). DexCap uses portable motion capture and imitation learning to connect human hand motion to dexterous robot-hand learning (Wang et al., 2024a). Human2Robot, Data Analogies, Polybot, and Scaling Cross-Embodied Learning use paired or multi-embodiment data to learn what remains stable and what changes across bodies (Xie et al., 2025; Yang et al., 2026; 2023b; Doshi et al., 2024). The interface and correspondence design determine which differences are absorbed in the data and which remain on the target robot. The same trajectory can reach a different contact point or exceed the reachable space when motion is transferred from a human hand to a robot hand or from one robot to another. Different controller responses also prevent the same command from producing the same force. Correspondence data therefore do not solve transfer by themselves. When contact and force are recorded as well, they can become valuable resources for learning what can be shared across embodiments and may support common representations of contact in the future.

Learning correspondence across embodiments addresses body differences more directly than the other two research directions. Task progress and object motion can communicate a goal without matching low-level commands, while morphology-aware methods can explicitly account for kinematic structure. Physical contact and real controller response become increasingly important as the shared structure moves closer to execution. A small change in contact can change the result even when the trajectory looks similar. Deformable objects make the problem harder because their shape continues to change during execution. Delays and differences in low-level control also change the motion produced by the same action representation. These factors vary across robots and are difficult to collect at scale. The central challenge is to share contact behavior and recovery across embodiments in addition to kinematic correspondence. Section 7 presents ways to record where such problems appear.

## 7 Reporting Work That Success Rate Alone Does Not Reveal

As Sections 4–6 show, the three research directions reduce different parts of the embodiment gap. Methods that broadly share semantics and perception still need to connect that information to motion on the target robot. Methods that share robot data or action representations must align the conditions of training with the way the target robot moves. Methods that learn correspondence across embodiments address problems closer to execution, while correction during contact and recovery after failure remain difficult to share.

Success rate is an important measure of whether a robot completed a task. The number alone does not show how much new data were collected, how often the real-robot setup was adjusted, or how often a person stopped an unsafe motion before that performance was achieved. Researchers, engineers, and students often

carry out this work in research laboratories, while system integrators carry it out in industrial deployment. Making this work visible is a necessary step toward moving robot foundation models from research results toward practical use.

This section proposes reporting practices that let readers follow the adaptation process on a target robot. The report card records the path from the source system to final evaluation. An Embodiment Adaptation Curve relates performance to a stated amount of adaptation work. A record of failure causes identifies the stage at which the system failed. Together, these reports allow readers to see how a reported success rate was obtained.

Table 1 is designed so that readers can follow the process from the source robot to final evaluation. The *item* column names a stage in that process. The *what to report* column specifies the facts that should be recorded at that stage. The *why it matters* column explains what a reader can infer from those facts. The source robot or source setting, the target robot, and the shared structure establish what changed across transfer and what part of the model, representation, or skill was reused. Without these conditions, a reader cannot tell whether a result reflects transfer between substantially different embodiments or movement between closely related robots.

The modified components, target-robot data, and model updates show where the system was changed and how much target-robot experience was used. A method that adjusts only an output head reuses a different portion of the original system from a method that updates the full model, even when their final success rates are similar. Data collected on the target robot are also part of the resources used to obtain the result and should be reported as such. The target-robot data item records what was collected and used for adaptation. The real-robot operation item records the physical effort needed to collect or use that data, including robot time, resets, and reruns. Calibration and setup, real-robot operation, and safety, intervention, and recovery record the work required to make a trained model ready for evaluation on physical hardware. Aligning the camera with the robot, repeating trials, and stopping or recovering the robot can affect deployment effort independently of the size of the model update. These activities are difficult to infer from success rate, so the report card treats them as separate items.

Finally, evaluation rollouts and failure causes distinguish adaptation from performance evidence. Evaluation rollouts show that final performance was measured under conditions that were not used to adjust the system. Failure causes help readers determine whether a problem arose in planning or perception, in connection to the target robot, or during contact-rich execution. Trials used to improve the system should be reported separately from trials used to measure final performance, because mixing them makes the meaning of the success rate unclear.

Table 1: Minimum report card for cross-embodiment results. The card helps readers trace work required on the target robot that success rate alone does not reveal.

| Item | What to report | Why it matters |
|---|---|---|
| Source robot(s) / source setting | Robot(s), sensors, control system, pretraining setting | Establishes the source conditions |
| Target robot | Body, sensors, task environment | Establishes the execution conditions |
| Shared structure | Plans, representations, skills | Shows what is reused |
| Modified components | Head, adapter, decoder, control system | Shows what is changed for the target robot |
| Target-robot data | Demonstrations, adaptation trials, recovery data | Shows experience used for adaptation |
| Model updates | Frozen components, adapter/LoRA, full fine-tuning | Shows the scope and cost of updating |
| Calibration and setup | Coordinate frames, cameras, gripper, control rate | Shows preparation before deployment |
| Real-robot operation | Adaptation trials, resets, robot time, reruns | Shows physical work before evaluation |
| Safety, intervention, and recovery | Stops, unsafe contact, interventions, recovery | Shows safety and recovery work |
| Evaluation rollouts | Held-out episodes, tasks, seeds, success criteria | Provides performance evidence |
| Failure causes | Semantics/perception, data, correspondence, execution | Shows where problems remain |

Table 2 applies the report card to representative cross-embodiment claims. Its purpose is to illustrate which items can be identified from public papers and which remain unclear. We use N/R when an item could not be identified from the main paper or publicly available supplementary material.

Filling in Table 2 separates information that can be identified from information that remains unavailable in current reports. Among the nine method groups, counts for safety, intervention, and recovery could not be identified from the main papers or public supplementary materials for seven groups. Setup or calibration conditions could not be identified in sufficient detail for three groups. Many papers do not state how the target robot was prepared, how often people intervened during evaluation, or how the system recovered after failure. Without this information, readers cannot tell whether a high success rate arose mainly from autonomous model capability or from careful adjustment and operation by researchers.

This reporting gap is consistent with the empty Safety / Recovery row in Figure 2. Safe stopping and recovery after failure are the final support when physical execution breaks down. They are essential for sustained operation in the real world. A high success rate does not establish the autonomy or practical value of a system when readers cannot see how much of this work was carried out by people or surrounding infrastructure. Safety and recovery should therefore be treated as central research problems for turning RFMs into systems that can operate continuously. They deserve a central place in model and system design as well as in evaluation.

The report card also shows where further research or engineering may be needed and makes differences in deployment conditions easier to examine when methods report similar success rates. Papers may not be able to disclose every item at the same level of detail. Stating which items were not reported still helps readers judge the limits of a comparison.

Table 2: Examples of applying the report card to representative cross-embodiment claims (Open X-Embodiment Collaboration et al., 2023; Octo Model Team et al., 2024; Kim et al., 2025c;b; Zhang et al., 2025b; Bousmalis et al., 2024; Black et al., 2024; Physical Intelligence et al., 2025; Bharadhwaj et al., 2024; Zhi et al., 2026; Sferrazza et al., 2024; Patel and Song, 2025; Zheng et al., 2026a; Zhang et al., 2026; Wi et al., 2026; Huang et al., 2025). N/R denotes information that was not identifiable from the main paper or publicly available supplementary material. Rows that combine related systems summarize reporting at the group level and do not imply identical placements in Figure 2.

| System | Shared structure and connection to target robot | Target data / model update | Setup / operation | Safety / intervention / recovery | Visible work on target robot |
|---|---|---|---|---|---|
| RT-X / OXE | Multi-robot data → robot-specific actions | Large mixed dataset; shared policy | Data and control conventions | Resets and safety mostly N/R | Differences across robots and adaptation costs remain visible |
| Octo | Generalist policy → target interface | Target data and adaptation | Controller alignment | Intervention and recovery N/R | Connection to the target robot remains |
| OpenVLA | 7B VLA + action tokens → FT/LoRA | 10–150 demonstrations; full FT or LoRA | Control rate partly reported | Qualitative recovery; counts N/R | Decoder and control-rate work |
| RoboCat / $\pi_0$ | Generalist backbone → robot/task adaptation | RoboCat: 100–1000 examples; $\pi_0$: FT data | Setup N/R | Safety and resets N/R | Zero-shot and fine-tuned regimes are mixed |
| Track2Act | Point tracks and object motion → residual policy | 400 Spot teleoperation trajectories; residual-policy BC | Depth and transform fitting | Failure modes reported; counts N/R | A target-robot residual policy is still required |
| MOTIF | Action motifs → few-shot transfer | 1–50 shots; motif-conditioned policy | Setup largely N/R | Operation and safety N/R | Operating cost remains a reporting question |
| Body Transformer / GET-Zero | Body graph → embodiment-conditioned policy | Low or zero target data; frozen/generated policy | Body specification | Contact and recovery outside the model | Kinematics and contact remain separate problems |
| X-VLA / RodriNet | Body prompt or kinematic prior → action generation | Target adaptation varies | Body metadata; calibration N/R | Touch, force, and recovery remain | Work remains after morphology is modeled |
| TactAlign / Tactile-VLA | Tactile and force cues → tactile/VLA channel | Sensor and task data; tactile module | Sensor placement and calibration | Contact reported; safety counts N/R | Contact information adds sensor-related work |

## 7.1 Recording adaptation work with Embodiment Adaptation Curves

A paper may not be able to report every item in full detail. We therefore introduce the *Embodiment Adaptation Curve (EAC)* as a compact way to relate performance to the amount of adaptation work performed on a target robot. A single final success rate does not show whether performance was obtained with little work or after extensive additional data collection and real-robot adjustment. It also does not show whether performance continues to improve as adaptation increases or reaches a plateau early.

An EAC places one comparable measure of adaptation effort on the horizontal axis and shows how performance changes as that effort increases. When the number of target-robot demonstrations is used, for example, the curve distinguishes a method that becomes capable with a few demonstrations from one that requires many. The same idea can be applied to real-robot trials or the number of human interventions. An EAC selects one measure that can be compared across the reported conditions and lets readers see how that work is related to performance.

A study that reports performance at several adaptation levels can produce a full curve. MOTIF is a convenient example because it reports few-shot cross-embodiment transfer for several numbers of demonstrations (Zhi et al., 2026). When a study does not report performance across several levels of adaptation effort, it can still provide an endpoint illustration. Track2Act compares open-loop execution with a residual policy trained on 400 target-robot teleoperation trajectories (Bharadhwaj et al., 2024).

Figure 4 illustrates the idea through a conceptual curve in panel (a), the relation between target demonstrations and performance for MOTIF in panel (b), and an endpoint comparison for Track2Act in panel (c). The residual-policy endpoint in panel (c) uses 400 target-robot teleoperation trajectories. The values were transcribed from MOTIF arXiv:2602.13764v1, Table 3, and Track2Act arXiv:2405.01527v2 / ECCV 2024,

Table 2. Panel (c) is an endpoint example related to the EAC idea rather than a quantitative adaptation curve. Track2Act reports two execution conditions, but it does not report performance at several levels of adaptation effort, so a full curve cannot be reconstructed from the published results. We include this comparison because many studies reviewed here report only one adaptation setting or a small number of endpoints. The information needed to draw a full EAC is therefore often unavailable.

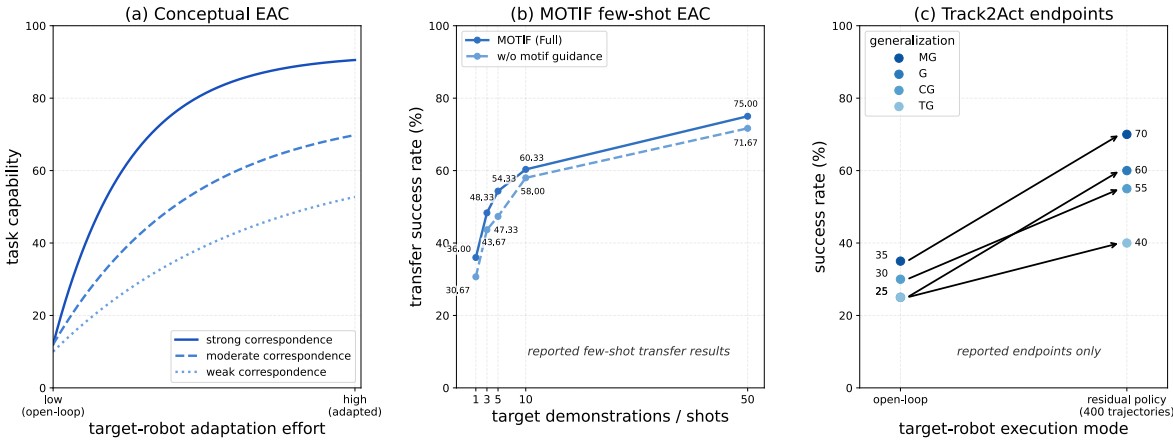

Figure 4: Embodiment Adaptation Curves (EACs). An EAC shows how performance changes with a reported amount of work on the target robot. The horizontal axis can use demonstrations, real-robot trials, human interventions, or another comparable measure of adaptation effort. Panel (a) gives a conceptual example, while panel (b) reproduces the few-shot transfer results reported for MOTIF (Zhi et al., 2026) in arXiv:2602.13764v1, Table 3. Panel (c) uses the values reported for Track2Act (Bharadhwaj et al., 2024) in arXiv:2405.01527v2 / ECCV 2024, Table 2, and compares open-loop execution with a residual policy trained on 400 target-robot teleoperation trajectories. It is an endpoint illustration related to the EAC idea rather than a quantitative adaptation curve. MG/G/CG/TG denote Mild, Standard, Compositional, and Type Generalization in Track2Act.

## 7.2 Organizing failure causes

Recording failure causes helps identify whether a problem arose in planning, data connection, correspondence across embodiments, or physical execution. A failure rate without this distinction does not show whether the model should be improved or whether the connection to the target robot should be revised.

For reporting failure causes, we use four broad groups that connect the three research directions to final execution on the target robot. The first group covers failures in semantics and perception: the instruction may be misunderstood or the object may be misidentified, leading the system to choose the wrong goal. The second group covers robot data and interfaces: an action representation from the training data may not match the interface of the target robot and may fail to produce the intended motion. The third group covers correspondence across embodiments: a target position may be unreachable, or an object motion may not be reproducible with the new body. The final group covers execution on the target robot, including unstable contact, unsafe motion, and failure to resume after stopping.

For example, if a model predicts the correct object flow but the gripper slips during execution, the primary failure lies in execution on the target robot. A secondary cause may be that the cross-embodiment correspondence did not account for the difference in contact. An affordance point outside the reachable workspace can be read as a correspondence failure. When an action representation from a VLA policy does not match the controller rate or action decoder, the failure can be recorded under robot data and interfaces.

Morphology-aware models also illustrate the distinction. A system such as RodriNet can reduce problems in joint motion or kinematics, while failures in contact and recovery can remain. Representing the body structure does not by itself make contact stable or provide a way to recover from failure.

Methods that use touch or force make these failures observable in another form. They record a contact failure as a tactile or force signal that can be used to understand or align execution. Such records are valuable for explaining failure in contact-rich tasks. The observation process itself still varies across robots and tasks. A study must decide what a sensor measures and when a signal should be treated as an early sign of failure. As RFMs become more multimodal, tactile, force, sound, and vision data recorded during failed execution can help explain failure causes alongside successful trajectories.

## 8 Toward Scaling That Reduces the Embodiment Gap

Section 7 presented ways to make work on the target robot visible in a paper. Once that work is reported, the next question is how research can reduce it when a model is moved to a new robot. The survey suggests four priorities for future work. Across these priorities, safe execution and recovery remain necessary parts of deployment.

First, data should preserve the process by which an action became executable, in addition to the final successful trajectory. Robot data should record how the target robot was adjusted and checked before execution succeeded. Contact-rich manipulation also benefits from records of how contact became stable and how the system recovered from failure. Data-collection systems therefore need to preserve changes and decisions made during adaptation, including real-robot adjustment, contact behavior, and safety checks. Such records can support reporting and may also become training data for future RFMs. A model that can learn why adjustment was needed, where execution failed, and how the system recovered may eventually learn part of the adaptation process itself.

Second, RFMs should be designed as systems that adapt to the target robot. Deployment involves work before and after a policy produces an action: aligning coordinate frames and control rates, correcting unstable contact, and stopping or recovering safely after failure. Future systems could estimate where additional adaptation is needed and assist with that adaptation. This would bring work currently performed by researchers and system integrators into the scope of the model and its surrounding system. Research is needed on the connection between a generalist policy and the control system that executes it. A higher-level agent could also select among policies and controllers and guide implementation. Such an agent would need to identify what is still mismatched on the target robot and help complete the necessary checks and adjustments. The research gap identified in Section 3.2 also calls for safe stopping and recovery to become central functions of generalist robot systems. A policy may contribute to these functions, while the final decision can also be handled by the controller or a supervisory agent. Failure detection, safe stopping, and recovery must be connected to the system that executes actions on the target robot. Developing that connection is an important research frontier.

Third, contact-rich execution should be moved toward structures that can be shared across robots with different bodies. Morphology-aware models can account for differences in joint structure and kinematics, as Section 6 showed. They still leave robot-specific questions about how contact is made, how much force is applied, and how motion is corrected after slipping. Deformable objects make these questions harder. Future data should therefore include touch, force, sound, and changes during contact in addition to vision and language. Sufficient contact and recovery data may support representations that can be reused across embodiments. It may also be useful to learn a rich multimodal representation from heavily instrumented robots and then estimate it on robots with fewer sensors. Handling different sensor configurations is a central part of the embodiment gap for generalist policies.

Fourth, evaluation should include the amount of work required to achieve success. The same success rate has a different meaning when one result uses a few demonstrations and light calibration while another uses many real-robot trials, resets, and human interventions. Future benchmarks and papers should therefore record the target-robot data, calibration, real-robot adjustment, safety checks, and recovery work required alongside success rate. The report card and EAC in Section 7 organize information needed for this comparison. As

such evidence accumulates, it can also help identify improvements needed in the first three research directions and guide the design of generalist policies and robot systems that learn adaptation and recovery as well as successful action.

## 9   Conclusion

This survey examined what can be reused across robots and what must still be prepared on a target robot when an RFM is deployed on a new embodiment. We called the gap between reusable structure and the work needed for physical execution the embodiment gap.

We organized recent work into three research directions: sharing semantics and perception, sharing robot data and interfaces, and learning correspondence across embodiments. These directions have expanded what can be shared. Substantial work still remains in turning shared structure into stable execution on the target robot. Alignment with the target controller, stable contact with objects, and safe recovery after failure are especially easy to miss when papers report only success rate.

We therefore proposed that papers report the work required to obtain a success rate. The report card and Embodiment Adaptation Curve in Section 7 provide ways to show this work in a paper.

Future scaling in robot learning must address physical interaction through a robot body as well as knowledge from language and vision. Training data can expand from successful actions to the process of adaptation, failure, and recovery that led to success. The embodiment-gap analysis in this survey provides a starting point. It also suggests that generalist policies should be connected more closely to control, safety, and recovery systems, and that contact and force should become multimodal shared structures. Progress in these directions can move RFMs from policies that output actions toward general robot systems that adapt those actions to the target robot during execution.

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

# A    Scope, Literature Selection, and Boundary Cases

## A.1    Scope of the survey

This survey focuses mainly on RFMs for manipulation, especially VLA policies and generalist robot policies. Manipulation makes the embodiment gap easy to see because differences in reach, grasping, contact, end effectors, sensors, controllers, and resets strongly affect whether a shared model or representation can run on a real robot.

The same idea can be extended to locomotion and navigation. In those settings, the work that remains is more likely to involve terrain, dynamics, localization, and long-term autonomy than object contact or grasping. We use manipulation as the main setting and discuss adjacent embodied-AI work when it helps explain reporting or evaluation.

The scope includes mechanisms around the model that help a shared model run on a target robot. We therefore consider data-collection methods and execution interfaces, methods for handling body differences, ways to observe contact, and evaluation systems. Considering these components together makes it possible to examine model reuse and the process of real-robot deployment as parts of the same problem.

## A.2    Literature-selection strategy

We constructed the literature set as a scoping survey organized around the questions of what is shared across robots and where work remains on the target robot. We reviewed recent surveys, conference and journal papers, OpenReview and arXiv records, and documentation for datasets and benchmarks. The search covered RFMs and VLA policies as well as multi-robot data, cross-embodiment transfer, teleoperation, retargeting, object flow, latent actions, morphology-aware control, touch and force, and evaluation. Most of the included work was available by early 2026. Earlier foundational studies were included when they were needed to explain a recent direction.

A work was included when it met at least one of four conditions. It introduced a widely referenced RFM or generalist robot policy; defined cross-embodiment data, interfaces, schemas, or benchmarks; proposed an explicit correspondence among bodies, objects, actions, or sensors; or made work on the target robot visible through evaluation or reporting.

Recent preprints and OpenReview submissions are included when they show an emerging research direction. The central claims of the survey rely, where possible, on published or accepted work and on widely used systems, datasets, or benchmarks. Unreviewed and under-review work is used as supporting evidence for the direction of the field. For recent references, we checked the public status and cited version as of July 14, 2026, and distinguish accepted papers from submissions and preprints in the bibliography.

## A.3    Boundary cases and inclusion decisions

Table 3 summarizes cases whose inclusion may be unclear. The table explains why each case is within the scope of this survey; it does not force the literature into mutually exclusive categories.

Table 3: Boundary cases and inclusion decisions.

| Boundary case | Why the case is near the boundary | Treatment in this survey |
|---|---|---|
| Body Transformer | This policy uses body structure and sits outside the usual VLA definition. | Included as morphology-aware sharing of actions and skills. |
| GET-Zero | It generates policies conditioned on a body graph. | Included as a morphology-aware method. |
| RodriNet | This kinematic mechanism lies outside a narrow definition of an RFM. | Included as a correspondence mechanism close to execution. |
| Tactile and force methods | These methods may lie outside VLA and directly address contact information. | Included because they make contact-rich execution visible. |
| Common APIs and control interfaces | These infrastructure components determine whether shared structure can be executed. | Included as connections needed to run shared structure on a target robot. |
| Object flow and point tracking | They can be read as visual cues or as correspondence across embodiments. | Included as methods that share changes in objects or scenes. |
| Latent actions | They are not low-level commands themselves. | Included as methods that share an abstraction of action. |
| Evaluation frameworks | They are not models. | Included because they can make work behind success rate visible. |

## B   Relation to Adjacent Concepts

Table 4 clarifies the relation between the embodiment gap and adjacent concepts. The deciding question is whether additional work is required to run transferred structure on a target robot after the embodiment changes.

Table 4: Relation between adjacent concepts and the embodiment gap.

| Concept | Example outside the main scope | Example treated as part of the embodiment gap |
|---|---|---|
| Domain shift | Only lighting, background, or object appearance changes on the same robot. | A different robot changes appearance together with reach, grasp, or control conditions. |
| Task adaptation | The same robot learns a new procedure or task. | The same goal requires a different grasp or motion on another body. |
| Sim-to-real | The same robot is transferred from simulation to hardware and only the physical domain changes. | Sim-to-real transfer is combined with transfer to a different robot embodiment, together with changes in the control interface or contact conditions. |
| System integration | General wiring, configuration, and implementation. | Calibration, control connection, and safety checks needed to run a shared policy or representation on a new robot. |
| Action-space mismatch | Only the output format differs. | The same action representation produces different contact or recovery problems on another robot. |
| Transfer learning | General reuse across domains or tasks. | Transferred structure must be made executable again under the body and control system of the target robot. |

Embodied cognition, ecological psychology, affordance theory, morphological computation, and developmental robotics have long situated action possibilities in the relation between a body and its environment (Wilson, 2002; Gibson, 1979; Pfeifer and Bongard, 2007; Jamone et al., 2018; Müller and Hoffmann, 2017; Chemero, 2003; Lungarella et al., 2003; Zech et al., 2017; Pfeifer et al., 2007). We cite this literature as background for the idea that body differences affect executability. A comprehensive review of embodied cognition is outside the scope of this survey.

## C   Placement Rules and Rationales for Figure 2

Each placement in Figure 2 selects one primary answer to two questions: what is shared across robots, and where work primarily remains on the target robot. The map does not rank methods or estimate the amount of work. It is a qualitative way to compare a method's central design with the execution problem that remains. The rules below are used when a method spans several stages.

### C.1 Definitions of the axes and boundary rules

The horizontal axis divides shared structure into five stages according to its distance from execution on the target robot. Table 5 explains how each stage is interpreted.

Table 5: Stages of shared structure used on the horizontal axis.

| Stage | What is primarily shared | Interpretation for placement |
|---|---|---|
| Semantic / Task | Goals, instructions, plans | The desired outcome is shared, while the physical action remains unspecified. |
| Perception / Affordance | Cues about objects and places for action | The target or location is known, but it must be converted into motion on the target robot. |
| Object Interaction | How manipulation should change an object or scene | The desired change is shared, while the action and contact that produce it depend on the embodiment. |
| Action / Skill | Latent actions, action tokens, skills, trajectories | Action structure close to execution is shared, but it must be connected to the target robot's action space. |
| Morphology-Aware Sharing | Body graphs, kinematics, body-specific conditions | Body differences are represented inside the policy to generate actions closer to the target robot. |

When shared structure spans several stages, we select the structure that the paper or method group presents as its central contribution to transfer across embodiments. A method that mainly identifies where to act is placed under Perception / Affordance. A method that mainly represents how an object or scene should change is placed under Object Interaction. The horizontal axis has no dedicated category for data infrastructure. We therefore place data-infrastructure groups under Action / Skill when trajectories or actions are reused as common training resources, and mark this choice as forced in Table 7.

The vertical axis divides the work required on a target robot into four stages. Table 6 explains each stage.

Table 6: Stages at which work remains on the target robot, used on the vertical axis.

| Stage | Work that primarily remains |
|---|---|
| Skill / API Connection | Connect plans or meaning to skills, functions, or planners that the target robot can actually call. |
| Calibration / Control Interface Alignment | Convert model outputs into commands that have the intended meaning in the action space and controller of the target robot. |
| Contact / Force Execution | Turn a spatially valid motion into physical contact, such as grasping or insertion, and correct deviations during contact. |
| Safety / Recovery | Stop safely after failures or disturbances and continue through retrying or recovery. |

When both control-interface alignment and contact-rich execution remain, we apply the following priority rule. Calibration / Control Interface Alignment is selected when a robot-specific action decoder, execution policy, or additional target-robot training is still required. Contact / Force Execution is selected when the method already maps its representation to target-robot actions and the remaining work is explicitly associated with grasping, force, friction, contact dynamics, or closed-loop physical correction. The Safety / Recovery stage is not selected merely because a method involves contact. It is used when failure detection, retrying, safe stopping, recovery, or sustained operation is a central unresolved part of the method.

### C.2 Independent coding and adjudication

Three researchers independently coded the 21 method groups using the same initial codebook. Each coder selected one primary category on each axis. When several choices were plausible, the coder also recorded a secondary category, confidence, a boundary-case flag, and a short rationale. The independent records were preserved and kept separate from the adjudicated placements.

On the shared-structure axis, all three coders agreed on 16 groups (76.2%), the mean pairwise agreement was 84.1%, and Fleiss' $\kappa$ was 0.78. At least two coders agreed for all 21 groups. On the axis for where work remains, all three agreed on 10 groups (47.6%), the mean pairwise agreement was 63.5%, and $\kappa$ was 0.33. At least two coders agreed for 20 groups (95.2%). All three agreed on both coordinates for 7 groups (33.3%). Ten of the 11 disagreements on the vertical axis were between adjacent categories. XSkill / UniSkill was the only group split across three stages.

Majority decisions served as the starting point for final placement. For boundary cases, we reread the main claim of each paper and the work on the target robot that could be identified from public information, then reached an adjudicated placement. Secondary choices, confidence, and boundary flags recorded the decision process, but coders used them differently, so they were not included in the main agreement statistics. The check does not establish that the categorization is fully objective. It shows that the horizontal axis is comparatively reproducible and that judgments on the vertical axis concentrate around the boundary between calibration and contact. We added the priority rule above during adjudication after the independent check identified this boundary as the main source of disagreement. The adjudicated coordinates are stored in a machine-readable table that provides the placements used in Figure 2 and Table 7.

### C.3 Placement rationales for Figure 2

Table 7: Adjudicated placements and rationales for representative method groups in Figure 2.

| System / method group | Primary shared structure | Where work remains | Research direction | Placement rationale |
|---|---|---|---|---|
| SayCan | Semantic / Task | Skill / API Connection | Semantics and perception | Shares language-level task structure; the target robot still needs to connect each candidate to an executable skill. |
| RT-2 / PaLM-E | Semantic / Task | Skill / API Connection | Semantics and perception | Shares semantic and vision-language knowledge from web data; the target robot still needs to connect that knowledge to available actions or skills. |
| CLIPort / PerAct | Perception / Affordance | Calibration / Control Interface Alignment | Semantics and perception | Shares spatial cues about objects and places for action; the target robot still needs to align the output with motion generation and its control interface. |
| RT-Affordance / RoboPoint / VoxPoser | Perception / Affordance | Contact / Force Execution | Semantics and perception | Shares affordances or spatial cues for interaction; the target robot still needs to realize those cues as a stable grasp or contact. |
| VPP / FlowDreamer | Perception / Affordance | Calibration / Control Interface Alignment | Semantics and perception / cross-embodiment correspondence | Shares visual temporal change predicted from video or flow; the target robot still needs to convert it into executable actions and align them with its controller. |
| DROID / LeRobot / RoboMIND | Action / Skill | Calibration / Control Interface Alignment | Robot data and interfaces | Reuses robot experience and trajectories in common formats; the target robot still needs to map the action representation in the data to its own control system. |
| RT-X / OXE | Action / Skill | Calibration / Control Interface Alignment | Robot data and interfaces | Shares multi-robot action data and action-generating policies; the target robot still needs alignment of its action space and control interface. |
| Octo | Action / Skill | Calibration / Control Interface Alignment | Robot data and interfaces | Shares an action policy across multiple robots; the target robot still needs its inputs, outputs, and controller to be connected and adapted. |
| OpenVLA | Action / Skill | Calibration / Control Interface Alignment | Robot data and interfaces | Shares a VLA backbone and action generation; the target robot still needs adaptation of the action decoder and control rate. |
| BridgeData / RoboNet / RLDS | Action / Skill | Calibration / Control Interface Alignment | Robot data and interfaces | Shares trajectory data and recording schemas; the target robot still needs the recorded actions to be mapped to its real control system. |
| Track2Act | Object Interaction | Contact / Force Execution | Cross-embodiment correspondence | Shares desired object motion through point tracks from video; the target robot still needs to realize that motion through stable contact. |
| MOTIF | Action / Skill | Calibration / Control Interface Alignment | Cross-embodiment correspondence | Shares action motifs across embodiments; the target robot still needs additional learning from a few demonstrations to connect the representation to an execution policy. |

| System / method group | Primary shared structure | Where work remains | Research direction | Placement rationale |
|---|---|---|---|---|
| Cosmos Policy / DreamZero (WAM) | Object Interaction | Calibration / Control Interface Alignment | Robot data and interfaces / cross-embodiment correspondence | Shares temporal scene changes represented by video models; the target robot still needs to connect those predictions to action generation and its control interface. |
| Flow-based methods | Object Interaction | Contact / Force Execution | Cross-embodiment correspondence | Shares a desired flow of objects or scenes; the target robot still needs to produce that change through force and physical contact. |
| RoboCat / $\pi_0$ / $\pi_{0.5}$ | Action / Skill | Calibration / Control Interface Alignment | Robot data and interfaces / cross-embodiment correspondence | Shares action policies learned across robots; the target robot still needs additional data to adapt the policy to its body and controller. |
| UMI / ALOHA | Action / Skill | Contact / Force Execution | Cross-embodiment correspondence | Shares action data through a common operation interface; the target robot still needs to reproduce the recorded motion as stable contact. |
| XSkill / UniSkill | Action / Skill | Calibration / Control Interface Alignment | Cross-embodiment correspondence | Shares skill representations across embodiments; the target robot still needs a skill-conditioned policy and additional learning to convert the representation into actions. |
| Body Transformer | Morphology-Aware Sharing | Calibration / Control Interface Alignment | Cross-embodiment correspondence | Shares a body graph of sensors and actuators for policy computation; the target robot still needs the output to be aligned with its actual control conditions. |
| GET-Zero | Morphology-Aware Sharing | Contact / Force Execution | Cross-embodiment correspondence | Shares policies conditioned on a body graph and generates actions for unseen embodiments; the target robot still needs to handle force and slipping during contact. |
| X-VLA / RodriNet | Morphology-Aware Sharing | Contact / Force Execution | Cross-embodiment correspondence | Incorporates body-specific information or kinematic structure into action generation; the target robot still needs stable physical contact with objects. |
| TactAlign / Tactile-VLA / ForceMimic | Action / Skill | Contact / Force Execution | Cross-embodiment correspondence | Shares action representations that include touch or force; the target robot still needs contact states to be aligned across sensors and translated into stable physical action. |

## D  Compact Reporting Checklist

Table 8 distills the adaptation- and evaluation-related items from the report card into a compact checklist for future work.

Table 8: Compact reporting checklist.

| Item | Information to report when available |
|---|---|
| Target-robot data | Demonstrations, adaptation trials, failure and recovery data |
| Model updates | Frozen components, adapter/LoRA, decoder, full fine-tuning |
| Calibration and setup | Cameras, coordinate frames, gripper, control rate, workspace |
| Real-robot operation | Robot time, resets, reruns |
| Safety, intervention, and recovery | Stops, unsafe contact, manual or autonomous recovery |
| Evaluation rollouts | Held-out rollouts, tasks, seeds, success criteria |
| Failure causes | Primary and secondary causes |

Work used for adaptation should be reported separately from evaluation rollouts used to measure performance. Adaptation work includes trials, resets, and data on the target robot that were used to adjust or improve the system. Evaluation rollouts are performance evidence and should not be counted as adaptation work. Conducting the evaluation can still contribute to the practical cost of real-robot operation.

**Worked report-card example.**  Table 9 gives a partial worked example for OpenVLA, and Table 10 illustrates how failure causes can be organized. The OpenVLA example does not reevaluate the system. It shows what can be identified from public papers and which items remain N/R.

Table 9: Worked report-card example for OpenVLA.

| Item | Information identifiable from public materials | Reading through the embodiment gap |
|---|---|---|
| Source robot(s) / source setting | Pretraining on Open X-Embodiment and reported out-of-box settings | Starting point of the shared VLA backbone and action tokens |
| Target robot | Fine-tuning setting on Franka | Conditions for execution on the target robot |
| Shared structure | 7B VLA backbone and tokenized actions | Components reused across settings |
| Modified components | Full fine-tuning or LoRA adaptation | Components changed for the target robot |
| Target-robot data | 10–150 demonstrations per Franka task across seven tasks | Target-robot experience used for adaptation |
| Model updates | Full fine-tuning or LoRA; rank-32 LoRA trains a subset of parameters | Update cost is partly visible |
| Calibration and setup | Control rate partly reported; calibration metadata N/R | Setup work is only partly visible |
| Real-robot operation | Evaluation rollouts reported; adaptation trials, resets, and robot time N/R | Physical operating cost remains unclear |
| Safety, intervention, and recovery | Qualitative recovery examples; counts of stops or interventions N/R | Safety and recovery work remains a reporting question |
| Evaluation rollouts | Reported task rollouts and success rates | Performance evidence is visible |
| Failure causes | Not reported systematically | Opportunity to organize causes of remaining work |

Table 10: Examples of organizing failure causes.

| Observed failure | Primary cause | Secondary cause |
|---|---|---|
| Object flow is correct, but the gripper slips | Execution on the target robot | Correspondence across embodiments |
| The predicted interaction point is outside the reachable workspace | Correspondence across embodiments | Execution on the target robot |
| The action token does not match the control rate | Robot data and interfaces | Execution on the target robot |
| The task plan is correct, but no usable skill is available | Semantics and perception | Skill / API connection |

