# OpenReview forum: "The Embodiment Gap in Robot Foundation Models"
_TMLR — Under review for TMLR_

### Review · Reviewer_ZJTs · 2026-06-23

**Summary Of Contributions:**

This paper presents a conceptual framework for analyzing the transfer of knowledge and capabilities across different robot embodiments. It introduces the notion of an embodiment gap to describe the mismatch between reusable model components and the additional work required to deploy them on a specific target robot. The survey organizes prior work according to the type of structure being shared and the main stage at which embodiment-specific adaptation remains necessary. It further categorizes existing research into three overlapping areas: semantic knowledge, multi-robot physical data, and cross-embodiment correspondence. Beyond reviewing technical approaches, the paper proposes a reporting framework for documenting target-robot data, model updates, calibration, physical trials, interventions, recovery, and failure sources. It also introduces embodiment adaptation curves as a way to relate task performance to different forms of target-side adaptation effort.

**Audience:**

Yes

**Audience Explanation:**

The paper addresses a timely problem at the intersection of foundation models, robot learning, and cross-embodiment transfer, which is relevant to the TMLR community. Its proposed vocabulary and reporting framework may also be useful to researchers working on generalization, adaptation, evaluation, and deployment of learned robotic policies.

**Claims And Evidence:**

Yes

**Claims Explanation:**

The paper’s main claims are supported by a broad and relevant body of prior work, and the proposed framework is illustrated through representative systems, comparison tables, and concrete reporting examples.

**Requested Changes:**

The paper would benefit from a sharper operational definition of the embodiment gap. In particular, the authors should provide clearer inclusion and exclusion criteria that distinguish embodiment-specific residuals from more general forms of domain shift, task adaptation, system integration, or sim-to-real transfer. A small set of contrasting examples would make the scope of the concept more precise and improve its usefulness as an analytical framework.

---

> ### Author Response · Authors · 2026-07-13
>
> Thank you for pointing out the need for a clearer operational boundary. We are adding an explicit operational definition of the embodiment gap, together with inclusion and exclusion criteria and contrasting examples. These revisions will be placed prominently in Section 2, with more detailed boundary cases provided in the appendix. We plan to upload the revised manuscript and a detailed point-by-point response by July 18.

---

> > ### Author Response · Authors · 2026-07-18
> >
> > We thank the reviewer for pointing out that the central concept needed a clearer operational boundary.
> >
> > --Reviewer comment (summary): Provide a sharper operational definition of the embodiment gap and clearer inclusion and exclusion criteria relative to domain shift, task adaptation, system integration, and sim-to-real transfer.
> >
> > Section 2.1 now defines the embodiment gap as the gap that arises when models, representations, or data reused across robots must be converted into executable actions that match the body and control system of a target robot. The operational test is whether additional work is required on the target robot after transfer across robot embodiments.
> >
> > Section 2.2 adds contrasting examples. A change in lighting or object appearance on the same robot is treated mainly as domain shift. Learning a new procedure on the same robot is treated mainly as task adaptation. Sim-to-real transfer on the same robot embodiment is outside the main scope of this survey. These problems enter the embodiment gap when they are tied to transfer to a different robot body and require additional work on the target robot.
> >
> > The section also distinguishes general system integration from the calibration, control connection, and safety work needed to run a shared policy or representation on a new robot. A concrete end-effector example explains how an apparently shared action interface can still produce different physical behavior. Appendix B summarizes the inclusion and exclusion cases in Table 4.
> >
> > Revision location: Sections 2.1–2.2, Appendix B, and Table 4.

---

### Review · Reviewer_1tXP · 2026-06-26

**Summary Of Contributions:**

This is a scoping survey and position paper of evaluating robot foundation models from a novel lens. Rather than asking the usual "which model/benchmark scales best," it asks a narrower question: when a model that was trained to share structure across robots reaches a new body, how much work is still left to do on that specific robot before it can act? The paper names "this" leftover work as "the embodiment gap", and the quantity it wants the field to start reporting is the residual adaptation burden, meaning, the target-side demos, fine-tuning, calibration, real-robot operation, and safety/recovery effort spent before a success rate is even measurable.

As I understand, there are three main focuses of the paper:

**A vocabulary and a two-axis lens (Section 3)**: One axis is the shareability level, how far the shared structure sits from execution and successfully running. The other is the dominant residual locus where the leftover work mainly shows up. Figure 2 plots ~21 grouped systems on these two axes.

**Three "scaling directions" (Sections 4–6, Figure 3)**: semantic scale (reusing meaning/perception via foundation models), physical-data scale (making robot trajectories and interfaces reusable and traceable), and embodiment-correspondence scale (learning relations across bodies via object motion, latent actions, body conditioning, tactile/force).

**A reporting agenda (Section 7)**: A minimum report card (Table 1), a five-item burden profile, Embodiment Adaptation Curves (EACs, Figure 4) that plot capability against a stated budget such as demos or interventions, and a four-layer failure-attribution scheme (Section 7.2, Table 9).

**Strengths**: The lens of this paper is very current. They separate two questions that usually get collapsed into one: did the robot eventually succeed, and what did the target robot have to be put through before that success was even possible. In terms of proof,  N/R audit in Table 2 is the most convincing part of the paper, because it turns a vague complaint (reporting is incomplete) into something countable.

**Weaknesses**: The main figure (Figure 2) does not match the coding table that is supposed to define it (Table 6), in several specific and checkable ways. A few of the paper's own framing claims (abstract, Section 8) promise outcomes the paper argues for but does not demonstrate. Also, because a large share of the evidence base is very recent 2026 preprints, parts of the survey rest on sources whose claims are not yet stable. None of these is fatal, but the first one touches the core contribution.

**Audience:**

Yes

**Audience Explanation:**

Strong Yes.

- The topic fits TMLR well and is active in its readership. Several papers cited in this paper (RoboCat, ActionEQA, Perception Stitching) are themselves TMLR papers, so the audience is clearly there.

- The contribution is the kind TMLR accepts regardless of novelty: a reframing plus a concrete reporting tool.

- The N/R audit (Table 2) and the report-card template (Tables 1 and 7) are immediately usable by anyone writing or reviewing VLA papers for robotics, even if they don't adopt the full three-scales taxonomy.

- The paper also bridges the gap between research and practical application of robotic models and will be beneficial for the wider community

**Broader Impact Concerns:**

No separate ethics statement is needed.

**Claims And Evidence:**

Yes

**Claims Explanation:**

It is a weak Yes.

**Verified**

- The paper's empirical reconstructions hold up against the original sources:

     - OpenVLA (Table 8): seven Franka tasks at 10 to 150 demos each, LoRA rank 32, controller rates only partly reported. Matches.

     - Track2Act (Table 2, Figure 4c): about 400 teleoperated Spot trajectories, goal-conditioned residual policy, depth plus rigid-transform   fitting. Matches.

     - RoboCat (Table 2): adaptation with 100 to 1000 demonstrations. Matches.


- Section 4 system descriptions (CLIPort, PerAct, Perception Stitching, ActionEQA) are accurate.
This matters because the survey's representational claims are its core, and the worked examples behind the reporting agenda are not misread.

- The one strongly empirical claim also holds: the text says seven of nine rows in Table 2 are N/R in the safety column and three are N/R for setup or calibration, and both counts check out. So the claim that target-side work is real and goes unreported is genuinely evidenced. This is the solid empirical backbone.

**Concerns**
- Figure 2 is built from two coding rules (Appendix C): "primary shared representation" and "most visible residual." Both are judgment calls, and the paper says systems often span several levels, so ultimately, is it an opinion?

- World models are a clean example: Cosmos Policy and DreamZero can also be placed under physical-data scale just as defensible by the paper's own Section 5 definition. There is no fact of the matter, only a coding choice, right?

- There is no inter-coder check showing others would place systems the same way, no argument for why these axes beat alternatives (representation hierarchy, embodiment invariance, etc.), and no evidence the taxonomy is predictive rather than just descriptive. For example, something like "XX placement predicts YY kind of failure you'll hit while transferring ZZ model from embodiment A to B"

- The main figure contradicts its own coding table. Figure 2 disagrees with Table 6 in several checkable spots:

    - TactAlign and Tactile-VLA are plotted in the morphology column and recovery row, but coded under action/skill with a contact/force locus (the rationale even says morphology is not their primary structure).

    - Cosmos Policy and DreamZero are plotted near calibration but coded with a contact/force locus.

    - RT-Affordance, RoboPoint, and VoxPoser are plotted a row below CLIPort and PerAct despite sharing the same coded locus.

    - The recovery and safety axis category is labeled and populated but never assigned to any system in Table 6.

    - Figure 2 is qualitative, so these are consistency defects rather than false findings, but it is the main figure meant to demonstrate the second contribution, and it does not match the table that generates it.

- The paper fills in the report card once for OpenVLA and reconstructs two adaptation curves, but never runs the comparison set up in Section 2.3 (two systems with similar success rates that differ in burden).

**Requested Changes:**

1. **[critical] Make Figure 2 internally consistent.** Several placements contradict their own coded coordinates in Table 6. Also, the "recovery and safety" axis category is labeled and populated but never assigned to any system in Table 6.

2. **[critical] Show the taxonomy is more than one team's judgment, or reframe it.** The placements rest on subjective coding with no evidence others would reproduce them. Run an inter-coder check (two or three people independently placing the ~21 systems using the Appendix C rules; the Table 3 boundary cases are where they'll diverge).

3. **[critical] Justify why shareability × residual-locus beats alternatives.** For example, representation hierarchy or embodiment invariance, and state whether the categorization helps predict anything or is purely descriptive.

4. **[critical] Substantiate or soften the benefit claims.** The abstract and Section 8 promise safer recovery and easier comparison/reproduction, but the paper only fills in a template once (Table 8). Run the comparison set up in Section 2.3: two systems with similar success rates, both burden profiles filled in, showing the proposed framework of reporting, surfaces a difference that the success rate hides.

5. **[recommended] Add critical depth: comparison and limits.** The literature is largely described rather than compared, and coverage skews toward success stories. Add cross-method analysis (why morphology-aware methods still fail at contact, why action tokenization hasn't solved transfer, why diffusion policies still need heavy calibration) and a discussion of what fundamentally limits embodiment transfer (contact-rich manipulation, deformable objects, latency, controller mismatch), which connects naturally to Section 7.2. This would strengthen the motivation!

6. **[recommended] Fix citation and consistency details.** Figure 4 and Section 7.1 expand "CG" as "Combinatorial" where the Track2Act source says "Compositional." Confirm the MOTIF EAC values in Figure 4b against the cited preprint version and state which version. Either add the missing latent-action point on Figure 2 or drop it from Section 3.2.

---

> ### Author Response · Authors · 2026-07-13
>
> Thank you for the detailed and constructive review. We appreciate your positive assessment of the survey’s motivation and potential value, and we agree that the reliability and interpretation of Figure 2 are central to the paper. In the revision, we are addressing the concerns you raised as follows:
>
> 1. We are conducting a system-by-system audit of the placements and coding labels, and we will correct inconsistencies.
>
> 2. We will clarify that Figure 2 is a qualitative, descriptive map for organizing representative systems. We will also explain more explicitly why the two axes were selected and discuss what they do not capture.
>
> 3. We will conduct an independent coding check and report the procedure, initial agreement, major boundary cases, and how disagreements were resolved. We will also revise Appendix C so that the placement criteria and ambiguous cases are easier to inspect.
>
> 4. We will soften claims about practical benefits where they are not directly supported by the available evidence. We will also expand the critical discussion of the compared approaches, including concrete comparisons where the source papers provide sufficiently comparable evidence.
>
> 5. We will verify and correct the classification of Track2Act, the reported EAC value and cited version for MOTIF, and the consistency of the discussion of latent-action methods.
>
> We plan to upload a revised manuscript and a detailed point-by-point response by July 18. We will post a notification in this discussion once the revision is available.
>
> Thank you again for identifying these issues so precisely.

---

> ### Author Response · Authors · 2026-07-18
>
> We thank the reviewer for checking the cited systems carefully and for identifying concrete inconsistencies in Figure 2. The review led us to audit the map, document the placement process, explain the role and limits of the two axes more clearly, and deepen the cross-method analysis.
>
> --
> Reviewer comment 1 (summary): Make Figure 2 internally consistent with the placement table, including the Safety / Recovery row.
>
> We rechecked all 21 systems or method groups. The final placements are stored in one coordinate table, and the same coordinates are used in Figure 2 and the revised placement-rationale table, Table 7.
>
> We corrected the specific inconsistencies identified in the review: TactAlign / Tactile-VLA / ForceMimic is placed under Action / Skill and Contact / Force Execution. Cosmos Policy / DreamZero is placed under Object Interaction and Calibration / Control Interface Alignment. RT-Affordance / RoboPoint / VoxPoser is placed under Perception / Affordance and Contact / Force Execution.
>
> After adjudication, no method group had Safety / Recovery as its primary remaining stage. This does not mean that safety and recovery have been solved. Section 3.2 now explains that safe stopping and recovery have not yet become central research targets in the method groups covered by this survey. We retain Safety / Recovery on the vertical axis because it is necessary for sustained operation in the real world.
>
> Revision location: Section 3.2, Figure 2, Appendix C, and Table 7.
>
> --
> Reviewer comment 2 (summary): Show that the placements do not depend on one person’s judgment, or reframe the map accordingly.
>
> Three researchers independently coded the 21 method groups with a shared initial codebook. On the shared-structure axis, all three coders agreed on 16 groups (76.2%), the mean pairwise agreement was 84.1%, and Fleiss’ κ was 0.78. At least two coders agreed on every group.
>
> On the axis for where work remains, all three coders agreed on 10 groups (47.6%), the mean pairwise agreement was 63.5%, and κ was 0.33. At least two coders agreed on 20 groups. All three coders agreed on both coordinates for 7 groups. Ten of the 11 disagreements on the vertical axis were between adjacent categories. Most concerned the boundary between aligning a policy or control interface with the target robot and the contact-rich execution that follows. XSkill / UniSkill was the only group for which the three coders selected three different stages.
>
> We preserved the independent records and kept them separate from the adjudicated placements. Majority decisions served as the starting point. For boundary cases, we reread each paper’s main claim and the work on the target robot that could be identified from public materials. Because the independent check showed that the calibration–contact boundary caused most of the disagreement, we added a priority rule for that boundary during adjudication.
>
> We do not present these results as proof that the map is fully objective. The revised manuscript states that the horizontal axis was comparatively reproducible, while the vertical axis contains a real boundary that requires judgment.
>
> Revision location: Section 3.2 and Appendix C, especially “Independent coding and adjudication.”
>
> --
> Reviewer comment 3 (summary): Explain why the two-axis organization is useful relative to alternatives, and state whether the map is predictive or descriptive.
>
> We revised Section 3.1 to explain what the second axis adds. A representation hierarchy can distinguish high-level plans from action representations, but it does not show the work required to connect those representations to the target robot’s skills or controller. An analysis of what remains invariant across embodiments also does not show whether the system can establish contact or recover after failure.
>
> The two-axis map complements these views by adding the stage at which work remains during execution on the target robot. Figure 2 is qualitative and descriptive. It does not predict the amount of work, rank methods, or claim to predict transfer success.
>
> Section 3.2 also explains why the axes should remain separate. Shared structure and remaining work are related, but they do not have a one-to-one correspondence. OpenVLA and UMI / ALOHA both share actions or skills, yet the main work left on the target robot appears at different stages because the methods extend to different points in execution.
>
> Revision location: Sections 3.1–3.2 and the Figure 2 caption.

---

> > ### Author Response · Authors · 2026-07-18
> >
> > --Reviewer comment 4 (summary): Provide evidence for claims about comparison, reproducibility, safety, and recovery, or soften those claims.
> >
> > We chose the second option in the reviewer’s request and softened the claims. The revised Abstract and Introduction do not claim that the reporting framework has been shown to improve safety, recovery, reproducibility, or comparability. They state that the framework identifies work that should be checked alongside success rate and makes missing information visible.
> > We did not add a direct comparison between two systems evaluated on different tasks and under different reporting protocols, because such a comparison could mix the effect of the reporting framework with unrelated differences in the systems and evaluations. We therefore followed the reviewer’s alternative suggestion and reduced the strength of the benefit claims.
> >
> > We retained the empirical N/R audit in Table 2 and expanded the explanation of what it supports. Seven of the nine groups lacked reportable counts for safety, intervention, and recovery, and three lacked sufficient setup or calibration details. These findings show that success rate alone often leaves important work on the target robot unclear.
> >
> > Section 8 now presents safety and recovery as priorities for future system research, rather than as demonstrated outcomes of the reporting framework. Figure 4(c) is also described as an endpoint example related to the EAC idea. The revised text explains that many studies report only one adaptation setting or a small number of endpoints, so a full quantitative curve often cannot be reconstructed.
> >
> > Revision location: Abstract, Introduction, Section 7, Figure 4, and Section 8.
> >
> > --
> > Reviewer comment 5 (summary): Add a deeper analysis of why current methods still fail and what limits transfer across robot embodiments.
> >
> > We revised Sections 4–6 to connect each method’s design to the work that remains on the target robot. Section 4 explains why shared semantics and perception still require a connection to available skills, alignment with the target robot’s controller, and stable physical contact. It also explains how the physical meaning of an action token can change with the target controller.
> >
> > Section 5 explains why common data formats and interfaces do not guarantee the same physical behavior on different robots. For diffusion and trajectory-generating policies, we discuss how camera conditions, robot response, control timing, and connection to closed-loop control affect real-robot performance.
> >
> > Section 6 separates methods that handle kinematics or body structure from the remaining problems of contact, touch, and recovery. It also discusses why deformable objects, latency, and differences in low-level controllers make transfer harder. Section 7.2 then uses these observations to organize failure causes.
> >
> > We also added text at the beginning and end of each section to discuss the overall value and limitations of the approaches when viewed from the perspective of scaling.
> >
> > Revision location: Sections 4–6, Section 7.2, and Section 8.
> >
> > --
> > Reviewer comment 6 (summary): Correct CG, verify the MOTIF values and cited version, and resolve the latent-action inconsistency.
> >
> > We corrected CG to “Compositional Generalization.”
> >
> > We checked the MOTIF values against arXiv:2602.13764v1, Table 3. This audit changed the 1-shot value for the full MOTIF model from 40.33 to 36.00. We checked the Track2Act values against arXiv:2405.01527v2 / ECCV 2024, Table 2, and corrected the G and CG open-loop values. The exact source versions and tables are now stated in Section 7.1 and the Figure 4 caption.
> >
> > We removed wording that implied that Figure 2 contained a separate point for latent-action methods. Those methods remain discussed in Section 6 as part of the broader research direction. We also checked the publication status and cited versions of recent 2026 sources as of July 14, 2026, and clarified their status in Appendix A and the references.
> > Revision location: Section 6, Section 7.1, Figure 4, Appendix A, and the reference list.

---

### Review · Reviewer_g1co · 2026-07-05

**Summary Of Contributions:**

This survey deals with the idea of an "embodiment gap" in learning-based robotics, that is the ability (or lack thereof) to transfer the same method to another physical robot. The work organizes recent literature in three main sets, which either (i) only share high-level structures, or also shares lower-level components, either implicitly (ii) or explicitly (iii). These three directions are dissected in detail in map (Figure 2). After reporting on existing works, the authors propose a system for reporting and describing new works, including detailed report cards, curves evaluating performance at different adaptation efforts (EACs), and recommending disclosure of failure causes. Finally, the authors highlight promising directions for future works, both practical and more open ended.

**Audience:**

No

**Audience Explanation:**

My main issue with this paper is that it appears to be strongly LLM-generated. While the authors acknowledge LLMs were used for polishing purposes, the overall language is needlessly complex and hard to parse. This is precisely what a survey should avoid. In the current presentation, I am not convinced I would read this survey instead of querying an LLM to provide a similar exposition. A few examples:
- The choice of words is unnecessarily complicated. Is there a good reason for calling something "dominant residual locus"?
- There are recurring LLM-sounding constructs, e.g. "not this, but this":  "These surveys map the landscape; this survey asks a different question: not which capability, module, or benchmark is present, but what shared structure transfers across bodies and what target-body burden remains. Methodologically, this paper is a lens-driven scoping survey rather than an exhaustive systematic review."
- Some sentences overuse technical terms at no added value: "This section proceeds through four layers–report card, burden profile, EACs, and failure attribution–that turn success rate into a traceable target-body pathway.".
- Short, sharp sentences can be found in several sections, e.g. "The concrete implication is reporting. Success rate becomes most useful when paired with the pathway that produced it: target data, model updates, calibration and setup, real-robot operation, safety events, recovery, and failure attribution."

**Claims And Evidence:**

Yes

**Claims Explanation:**

- To the best of my knowledge, the description of the literature is both accurate and well organized. I find the distinction in three directions to be particularly clean and informative.
- The survey focuses on a topic which has not been dealt with in detail so far.
- I am not convinced that a report cart such as that presented in Table 1 / Appendix D is sufficiently clear. It presents a large amount of items with minimal explanation.

**Requested Changes:**

- The language in which the context is presented needs to be reworked in my opinion, as stated in the section above.
- Figure 1 is in .png format: a vector graphic would be preferable. IIt would also be great to explain it more in detail. At the moment, it presents a list of terms which are not introduced properly. For instance, why are representations split into three categories?

---

> ### Author Response · Authors · 2026-07-13
>
> Thank you for your candid and helpful comments on the readability and presentation of the manuscript. We are revising the manuscript throughout—not only the passages cited in the review—to make the prose more direct and easier to follow. We are also adding clearer introductions to the figures and clarifying how each section contributes to the paper’s overall argument. In addition, we are replacing Figure 1 with a vector graphic and aligning its terminology with Section 3 and Figure 2. We plan to upload the revised manuscript and a detailed point-by-point response by July 18.

---

> > ### Author Response · Authors · 2026-07-18
> >
> > We thank the reviewer for the candid comments on readability and presentation. We understood the review as a request to revise the manuscript as a whole, rather than to edit only the quoted examples.
> >
> > --
> > Reviewer comment 1 (summary): The manuscript uses unnecessarily complex and LLM-like language, including noun-heavy terms, repeated contrast constructions, compressed lists, and short slogan-like sentences.
> >
> > We rewrote the English manuscript throughout in plain language. Phrases such as “dominant residual locus,” “residual adaptation burden,” “target-body pathway,” and “lens-driven scoping survey” were removed or replaced with ordinary descriptions. We also reduced repeated contrast constructions and replaced compressed technical lists with sentences that explain why a particular kind of work remains on the target robot.
> >
> > We revised the paragraph structure in Sections 4–6 so that an idea, the representative methods, and the explanation of their limits appear in one continuous argument when they belong together. Some passages became slightly longer, but we kept the detail needed to understand the methods and their limitations.
> >
> > Section 8 was also thoroughly revised. Based on the analysis developed in the preceding sections, the revised version presents concrete research challenges for robot foundation models.
> >
> > We also added signposting at the beginning and end of each section to make the progression of the manuscript easier to follow.
> >
> > Revision location: The full manuscript, especially the Introduction, and Sections 4–8.
> >
> > --
> > Reviewer comment 2 (summary): Replace Figure 1 with a vector graphic and explain its concepts more clearly.
> >
> > We redrew Figure 1 as an editable vector graphic. Its terminology now matches Figure 2 and the main text. The left side shows structure that can be reused across robots, and the right side shows work that often remains before that structure can be executed on the target robot.
> >
> > Section 2.1 introduces the operational definition of the embodiment gap before the figure and explains its basic idea. Section 3.1 then explains why shared structure and remaining work are treated as separate axes in the qualitative map.
> >
> > Revision location: Section 2.1, Figure 1 and its caption, and Section 3.1.
> >
> > --
> > Reviewer comment 3 (summary): Table 1 and Appendix D contain many items, but the purpose of those items is not explained clearly enough.
> >
> > We expanded the text before Table 1 to explain why each group of items is included. The source and target settings show what changed across the transfer and what was reused. The next items record which parts of the system were modified, what data were collected on the target robot, and how the model was updated.
> >
> > The revised explanation then follows the process from preparing the system for real-robot evaluation to assessing its final performance. It shows what work was needed to make the system ready to run on the target robot and which trials were used to assess its final performance. It also explains the relationship between target-robot data and the physical work required to collect and use that data. These describe different parts of the adaptation process and are therefore reported separately.
> >
> > We rechecked Table 2 to identify which details could and could not be recovered from published reports. Appendix D is now presented as a Compact Reporting Checklist that summarizes the adaptation- and evaluation-related items in a form that can be used more easily.
> >
> > Revision location: Section 7, Tables 1–2, and Appendix D.